# Sustainable Tourism around Ecosystem Services: Application to a Case in Costa Rica Using Multi-Criteria Methods

**Juan Diego Araya** [1,2,*], **Ana Hernando** [1], **Rosario Tejera** [1] **and Javier Velázquez** [3]

1   Silvanet Research Group, E.T.S.I. Montes, Forestal y Medio Natural, Universidad Politécnica de Madrid, Ciudad Universitaria, 28040 Madrid, Spain
2   Sede Regional del Sur, Universidad de Costa Rica, Golfito 60701, Costa Rica
3   Faculty of Science and Arts, Catholic University of Avila, Calle de los Canteros, s/n, 05005 Avila, Spain
*   Correspondence: juandiego.araya@alumnos.upm.es or juan.arayavargas@ucr.ac.cr

**Abstract:** Tourism is an activity that generates important benefits; in the case of Costa Rica, it focuses its activity on natural areas, using the different ecosystem services (ES) and obtaining economic and social benefits. However, its ecological value can diminish, making it necessary to implement methods to estimate its sustainability. This paper proposes an evaluation of tourism around ES by applying the MIVES method (Value Integrated Model for Sustainability Evaluation), based on the multi-attribute utility theory, which implies that it transforms different variables into a 0 to 1 relationship, where the closer it is to 1, the more sustainable it is. For this purpose, it considered a decision tree, integrating environmental, economic, and social requirements, 6 criteria, and 13 indicators. The method was applied to two sites, Golfito and Jimenez in Costa Rica. It considered the following stages: (i) literature review and expert consultation, (ii) decision tree, (iii) assignment of weights, (iv) sustainability indexes, and (v) sensitivity analysis. The most sustainable site is Jiménez, obtaining an overall index of 0.40 compared to 0.25 for Golfito. The economic and social requirements are the best evaluated, while the environmental requirement was the worst evaluated for both sites. The proposed methodology can be extrapolated to other natural areas.

**Keywords:** ecosystem services; evaluation; sustainability; MIVES; tourism; decision-making





## 1. Introduction

Costa Rica is host to nearly 6% of the world's biodiversity in a territory of 51,100 km$^2$ [1], making it a megadiverse country [2]. Among the ecosystem services provided by natural spaces, those that are exploited through tourism stand out, mainly biodiversity and cultural resources providing essential services for the proper functioning of the ecosystem and society in general [3].

Tourism is an activity that generates economic, social, and also environmental benefits in terms of protection and conservation; however, it can also generate impacts if it is not managed in a sustainable manner. The relationship between humans and the environment is complex and highly variable; therefore, it is necessary to establish methods that contribute as verifying means to evaluate and analyze the interrelations between the environment and human beings. Different methodologies have been developed that seek to analyze, evaluate, and propose improvements in favor of sustainability; however, evaluating sustainability is not a simple process due to the number and diversity of variables that can influence it and the multiple elements that are interrelated with each other [4].

Evaluating sustainability has its foundations in the first conceptualizations of sustainable development, mainly those promulgated in the Brundland report "Our Common Future" [5], in which the economic model is questioned with respect to environmental sustainability, referencing the disparities between economic development and environmental conservation [6]. Through the IB it was possible to define sustainable development, understood as "development that meets current needs without compromising the ability of

future generations to meet theirs" [7], a definition that at the Earth Conference in Rio de Janeiro 1992 integrated what are considered the "three pillars" of sustainability, economic development, social equity, and environmental conservation, from a single perspective [8].

According to the IB, sustainable development is based on equity and social and environmental distribution, questioning models related to economic efficiency, encouraging debate on the true concept of sustainable development [9]. Sustainable development should be considered a multidimensional process in which the environmental dimension is oriented towards the balanced use of the natural environment, the economic dimension towards an equitable distribution of benefits, and the social dimension towards cohesion and shared progress [10].

Sustainability can be defined as "the capacity of a system to readjust or adapt its socio-ecological structures and interactions to possible disturbances and to persist without significant changes in its essential functions" [11]; it has to do with the capacity to maintain an activity indefinitely, establishing a line of compatibility between economic and social development and environmental protection [12]. Sustainability can be divided into two types: on the one hand, weak sustainability, which is characterized by a more economic orientation, and strong sustainability, which is based on an ecological principle [10]. In addition, it can be seen from a socioeconomic and management point of view [13], fostering the link between people and the social, economic, and environmental setting [14].

The main purpose of sustainability is to seek a balance between the economic, environmental, and social spheres; therefore, in the case of certain analyses and evaluations, if only one or two spheres are looked at, the perspective of the problem in question will not be comprehensive, it will only be partial [15]. Sustainability is identified when there is a stable, flexible, and resilient environment [16], which evidences the maintenance of the thresholds of a given ecosystem to enable regeneration over time [17].

At the tourism level, it is common to find sustainability and competitiveness in a similar way; however, the interaction between these two concepts rather than similarities can be seen as a partnership between the two and one that in practice is carried out as such; thus, for example, a tourist destination that offers a certain product with high standards of quality and sustainability can be very competitive because over a period of time it has made efforts to generate attachment to the product by the target consumers; consumers (tourists) can feel identification with the product and therefore want to buy it, generating loyalty [18]; it is here, where associated factors are determined to compete against other destinations that can offer something similar, that it has been demonstrated that the use of strategies linked to sustainable tourism, considering elements such as protection and conservation of ecosystems, mitigation of emissions and the fight against climate change, reduction in waste and pollution, plus green and environmentally responsible consumption [19], serve to create unique experiences through high quality products that contribute to make the destination more competitive [20].

Assessing sustainability implies maintaining the product offered over time, achieving stability without losing competitiveness with respect to others [21], and providing information for short- and long-term decision-making [22], and it is a decision-making strategy [23], so that actions can be defined to make the activity developed increasingly sustainable [24]. To assess sustainability, various methodologies have been applied in different fields of study, favoring the linking of processes to provide satisfactory solutions for diverse objectives depending on the field of application [25]. Some of the main and most used methods are indicator systems, generally quantitative [26], which require a great capacity to synthesize information, as well as the selection of sufficiently representative indicators that respond to the analysis process [27]. They are instruments that make it possible to specify in greater detail the reality of an area by observing the current status, monitoring possible changes, and at the same time promoting future actions [28], allowing the clarification of objectives and impacts so that changes can be made to achieve an ideal scenario [29].

Other tools used to evaluate sustainability applied to ecosystems include the sustainability barometer, which is based on evaluating the well-being of the ecosystem with respect to the environment and human well-being for the social part [30], and the ecological footprint, which estimates the consumption and supply of natural capital generated by a given activity [31]. Finally, life cycle assessment emphasizes measuring the potential environmental impacts that a given product may generate during its life cycle [32].

The evaluation of ES is generally associated with economic quantification or biophysical evaluation [33], which is why they have been considered as goods and services converted into economic goods [34]. Recent studies also highlight the valuation of ES from a social point of view, given the importance of evaluation from a non-monetary approach [35]; however, the analysis by incorporating the three requirements that make up sustainability [25] as a whole is not common, which limits the ability to obtain a more comprehensive view of the degree of sustainability of ES.

In the tourism activity, it is common to find evaluations using indicator systems, for example, obtaining indexes by means of a composite indicator, which allows to perform, through different calculations, individual indicators that represent the components of the concept being measured, providing a multidimensional evaluation [36]. Evaluations can also be made by weighting indicators, understanding that not all are equally important, and assigning higher weights to those indicators that are considered a priority according to expert criteria [37]. Some models that have been developed to estimate the social value of ES are the SolVES model (Social Values for Ecosystem Services Model), which provides quantitative indicators intended to show the priority of those stakeholders in the evaluation process [35]. Through the socio-cultural evaluation of ES, it is possible to include ecological valuations given that environmental management policies and strategies are promoted with greater effectiveness and better results [38].

Some sustainability indexes applied to ES in tourism include a sustainability index in natural areas based on value functions, which seeks to homogenize the values obtained from the different variables in adimensional values so that a sustainability value can be obtained [39]. The beach quality index [40], responds to a functional analysis by means of three sub-indices (natural function, protection, and recreational function) that provide the different ES provided in this case by the beach. Another index that has been implemented to evaluate the impact of land use change on terrestrial ES is the composite index, which is made up of a series of indicators considering chemical, physical, and biological variables to determine the effects of land use change [41] likewise, at the marine and coastal level, an evaluation was implemented through a set of indicators considering two main areas, the biophysical area and its link with the social dimension, obtaining a final sustainability index [42].

The evaluation of ES generally bases its analysis around one or two areas, whether from an economic, biophysical, or social quantification point of view [43]; therefore, this study aims to apply a methodology to evaluate tourism activity around ES by integrating the three pillars that make up sustainability. The research presented in this paper has two main objectives: (1) to obtain an index of sustainability of tourism activity around ES so that it can be applied in other natural areas, and (2) to apply the MIVES method in other fields of study to confirm its functionality as a tool for sustainability assessments in tourism activity and ES.

The research is novel because it opens new scenarios to assess sustainability through the application of a multi-attribute methodology that has only been implemented in the field of civil engineering, project management, and construction [44–46]. It also allowed the identification of weaknesses in how tourism activity is managed in the site of analysis, contributing advances to assess the sustainability evaluation processes clearly and objectively through reliable data that determine how distant the tourism destination is from reaching true sustainability scenarios.

## 2. Materials and Methods

There are several multi-criteria methodologies that have been implemented over time and applied in many fields of study since they facilitate in a more agile and practical way the choice of alternatives among a given set. In the study of sustainability, they are especially relevant because they allow for obtaining results considering a wide number of elements at environmental, economic, and social levels, allowing for clearer and more concrete decision-making [25].

Among the different multi-criteria methodologies that have been applied is the MIVES method (Integrated Value Model for Sustainability Evaluations); which is frequently used in structural and construction engineering [46]; however, it has been proven that it can be used in different areas, considering both products and services [47]. By means of MIVES, it is possible to analyze a wide set of alternatives, determine how sustainable they are, and make decisions based on these results. For this purpose, sustainability indexes are obtained from a decision-making tree previously shaped and weighted, as well as the application of value functions to normalize in the same unit the different values obtained according to the variables or indicators used [48]. The following sections will describe the application of the method in greater detail.

### 2.1. Study Area

The study was undertaken in the municipalities of Golfito and Puerto Jimenez, in the province of Puntarenas, Costa Rica. These are divided by the Golfo Dulce, sharing territories that make up the Osa Peninsula, which is located on the southern coast of the Pacific Ocean, with an area of approximately 1740 km$^2$. The study area is shown in Figure 1.

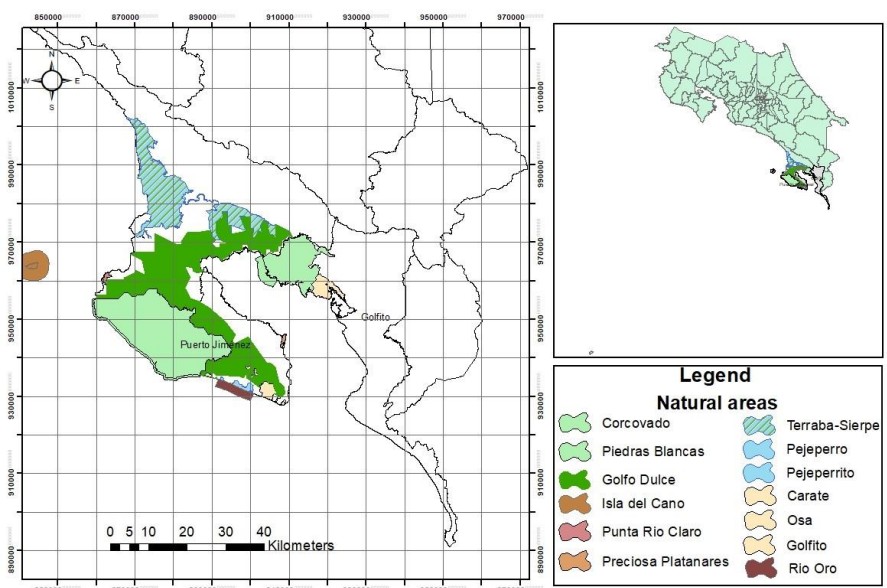

**Figure 1.** Natural areas around Jiménez and Golfito.

The study area is located in one of the most biodiverse areas in Costa Rica, considered a mega-diverse site, on which various economic activities are generated around natural attractions, with the presence of protected areas such as Corcovado National Park, the Golfo Dulce Forest Reserve, and the Golfo Dulce marine area.

### 2.2. Description of the MIVES Method

The methodology applied in the study is based on the author's master's thesis, in which the MIVES method was applied [49].

The integrated model of value and evaluation of sustainability, known by its acronym MIVES, is a tool that allows the application of multi-criteria decision-making processes in the field of sustainability. Its implementation initially occurred in the construction

sector [48]. However, given its versatility, the tool can be implemented in various areas of sustainability, from infrastructure evaluations to project management, as a decision-making technique [50]. The method can be used to evaluate both products and services, considering economic, social, and environmental requirements [51].

MIVES permits decision-makers to make decisions based on solid theoretical and practical foundations, allowing them to propose solutions to certain problems and thus contribute to sustainability [50]. The main problem faced by a decision-maker is the choice among several possible alternatives [52]. Therefore, by means of MIVES, each alternative can be evaluated by identifying those that can be more sustainable [53], obtaining a value index for each alternative proposed by means of a weighted sum of the valuations of the different criteria considered for the evaluation. The methodology is based on value analysis, which implies transforming variables of different types into a single unit [54].

In assessing sustainability in a comprehensive manner, at least the three main dimensions (environmental, economic, and social) must be considered [55], with the MIVES method being a tool that allows consideration of these dimensions. The MIVES structures a given problem or set of alternatives in a multicriteria analysis framework so that they can be prioritized according to the degree of importance through three levels, from the most general to the most specific, these being the requirements (pillars of sustainability), the criteria on which the evaluation will be based, and the indicators that will allow measuring and quantifying the degree of sustainability of each alternative under analysis [56].

The applicability of the method aims to prove that it is also functional in other fields of study; to date, its application has been mostly in studies related to civil engineering [45,57], project management [50], and construction [58]; however, given its flexibility, it can be implemented to evaluate both products and services [51]. In this particular case, tourism is the field of industry as an activity that integrates a diversity of products that are transmitted through a service [59], and therefore, the implementation of the method allows for extending the application scenarios, thus confirming the versatility of the model for different fields of study.

### 2.2.1. Decision Tree and Weight Evaluation

In the sustainability assessment process, the establishment of the decision tree is considered one of the most important steps. It will be the basis on which the analysis will be based with each of its requirements, criteria, and indicators [48]. In Table 1, the decision tree is detailed and consists of 3 requirements, 6 criteria, and 13 indicators.

**Table 1.** Decision tree and weight evaluation. The values marked in bold represent the weights assigned for the evaluation.

| Requirements | Criteria | Indicators |
|---|---|---|
| $R_1$ Environmental (38%) | $C_1$. Operation management (51%) | $I_1$ Environmental management plan and certifications (**55%**)<br>$I_2$ Wastewater management (**45%**) |
| | $C_2$. Climate Change and Biodiversity (49%) | $I_3$ $CO_2$ Emissions (**49%**)<br>$I_4$ Biodiversidad (**51%**) |
| $R_2$ Economic (31%) | $C_3$. Economic benefits and competitiveness (47%) | $I_5$ Income from tourism (**58%**)<br>$I_6$ Tourism declaration (**42%**) |
| | $C_4$. Supply chain of the activity (53%) | $I_7$ Local products and services (**63%**)<br>$I_8$ Benefits in the value chain (**37%**) |
| $R_3$ Social (31%) | $C_5$. Social impact (52%) | $I_9$ Jobs generated (**57%**)<br>$I_{10}$ Local tourism businesses (**43%**) |
| | $C_6$. Social perception (48%) | $I_{11}$ Belonging to the SE (**30%**)<br>$I_{12}$ Perception of the conservation of the ES (**30%**)<br>$I_{13}$ Satisfaction of the local population with how the ES are used in tourism activities (**40%**) |

The indicators were established through an initial literature review, considering as a base document the World Tourism Organization's guide of indicators [26], which proposes indicators to evaluate the sustainability of tourism destinations, and which can be adaptable to the context of each place. Figure 2 shows a diagram explaining the process of establishing indicators.

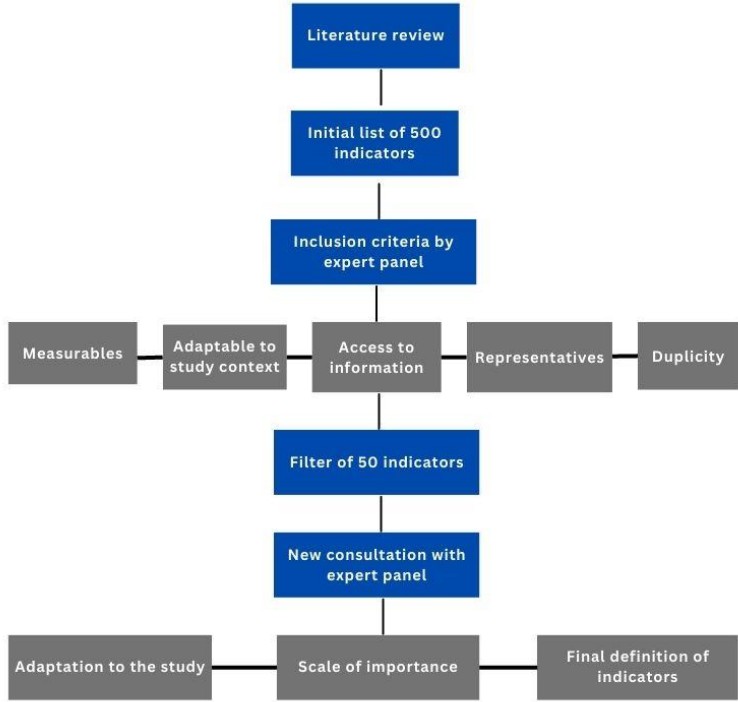

**Figure 2.** Explanatory diagram.

The database of 500 indicators was used as a starting point. Using inclusion criteria, a final list of 50 indicators was obtained, which were analyzed and evaluated with the panel of experts, resulting in a final list of 13 indicators for the sustainability assessment. The inclusion criteria were established through consultation with the panel of experts, considering five key aspects:

- Measurability: select those indicators that can be measurable or quantifiable.
- Adaptability to the study context: indicators that can be applicable to the study site considering its specific characteristics.
- Access to information: consider those indicators for which information is available or where the necessary methods can be used to collect the information.
- Representativeness: indicators that are as representative as possible of each area of application, whether environmental, economic, or social.
- Duplication: in sustainability assessments, it is common to find overlapping or duplicated indicators that require a large workload, so the purpose of each indicator was analyzed in detail to avoid duplication of information.

In sustainability evaluations using indicator systems, a range of 12 to 21 indicators is recommended, since a high number of indicators can mean difficulties in handling a large amount of information that could limit the scope of the study, and the values obtained from less important indicators can influence the results of the most important indicators [52], so priority was given to determining and selecting the most representative indicators in accordance with the context of the study. The 50 final indicators were subjected to a final evaluation by the panel of experts, for which a scale of importance was established from 1 to 3, where 1 is less important, 2 is moderately important, and 3 is very important. In the end, with all the evaluations, the percentage values of the choices made by the panel of experts were obtained, resulting in a final list of 13 indicators that were considered sufficiently representative to carry out the evaluation.

The conformation of the decision tree is shown below:

The environmental requirement ($R_1$) considered two criteria: operation management ($C_1$) and climate change and biodiversity ($C_2$). The first criterion includes two indicators defined as follows:

- $I_1$. Environmental management plan and certifications evaluate matters related to the environmental management of tourism establishments in terms of the protection and conservation of the ES they use for tourism activity.
- $I_2$. Wastewater management: considers the management of the establishments' wastewater and its different treatments.

The second criterion ($C_2$) also consists of two indicators:

- $I_3$. $CO_2$ Emissions: this evaluates the total $CO_2$ emissions emitted to the atmosphere as a product of transportation dedicated to tourism activity.
- $I_4$. Biodiversity considers the intensity of use by the tourist establishment and the use of biodiversity for its benefit, and if conservation and protection actions and policies are implemented to a greater or lesser extent.

The economic requirement ($R_2$) included two criteria: economic benefits and competitiveness ($C_3$) and the activity's supply chain ($C_4$). Criterion $C_3$ considers the evaluation of two elements: on the one hand, identifying and evidencing the economic benefits generated by the tourism activity in the destination in terms of the level of income received by workers in the activity, and on the other hand, identifying differentiating elements that contribute to the competitiveness of the destination:

- $I_5$. Income received from the tourism sector: this indicator seeks to estimate the range of income received by workers in the tourism destination so that the contributions of tourism to the local economy can be estimated.
- $I_6$. The tourism declaration: this is a type of certification implemented by the Costa Rican Tourism Institute (ICT), with the purpose of contributing to improving the quality of the tourism product offered to promote competitiveness. The purpose of this indicator is to measure the percentage of tourism establishments with respect to the total that has a tourism certification so that a measure of maximum or minimum satisfaction can be established according to the number of establishments with certification.

Criterion $C_4$ evaluates the productive chains generated by the tourism activity that, at the same time, receive a benefit. These are the ones that provide the necessary inputs for the activity, i.e., suppliers of services and local products. The following indicators will be used to evaluate this criterion:

- $I_7$. Local products and services: This measures the percentage of tourist establishments with respect to the total that purchase and consume products and services specific to the tourist destination. The destination's own products are those purchased in the local communities of the tourist destination, including agricultural products that supply restaurants and hotels, as well as handicrafts and products used for cleaning in the different tourist establishments. In the case of services, it has to do with transportation services, tour guide services, external cleaning services, professionals, maintenance, and mechanics, among others.
- $I_8$. Benefits in the value chain: the total percentage of local products and services that are purchased by tourism establishments, as a percentage of the total purchased by the establishment, and how much it represents at the operational level.

The social requirement ($R_3$) considered two criteria: ($C_5$) Social impact, and ($C_6$) Social perception. Criterion $C_5$ refers to the possible impacts perceived regarding the tourism activity in the destination, considering two key aspects: the generation of employment and the establishment of local tourism businesses by and for residents of the community.

- $I_9$. Jobs generated: evaluates the employability of local people in the tourist destination considering those jobs that are permanent full-time jobs and local jobs and the variability of employment due to the tourist season.

- $I_{10}$. Local tourism businesses: considers the number of tourism establishments in which the owners and direct investors are residents of the community where the tourism activity is developed; this indicator will allow the analysis of variables such as entrepreneurship and local business development.

Criterion $C_6$ estimates the perception of the resident population considering aspects such as belonging, conservation, and level of satisfaction with respect to the management of the ES in tourism activities. It contemplates the implementation of three indicators:

- $I_{11}$. Belonging to the ES: measures the perception of the local population regarding the sense of belonging and degree of importance of the ES for their community.
- $I_{12}$. Perception of the conservation of the ES: measures the perception of the local population regarding the conservation and protection of the ES and their use in tourism activities.
- $I_{13}$. Degree of satisfaction of the local population on how ES are used in tourism: measures the level of satisfaction of the residents with how ES are managed in tourism activities.

For the evaluation process, it is necessary to carry out a weighting of the decision tree through the application of criteria of twelve experts with experience in tourism and sustainable conservation; using a spreadsheet, they established the weights of the different elements, contemplating the environmental, economic, and social requirements. Given that in the end there will be great variability in the weightings, in order to assign the final weight with which the evaluation was carried out, the panel of experts was asked to establish the degree of certainty with which they were assessing the assignment of weights.

For this purpose, a scale from 1 to 3 was established, where 1 is defined as unsafe, and 3 as very safe; subsequently, with the assigned weights, it was calculated again taking as a reference the answers given by the experts applying the 3-point scale [47]. Thus, for example, if an expert assigns a weight of 30% to the environmental requirement with a response of 1 (unsafe), and another expert assigns a weight of 50% with a response of 3, the final weighted weight will be $(1 \times 30\% + 3 \times 50\%)/(1 + 3) = 45\%$. Table 3 shows the weights assigned to the decision tree by the expert consultation, considering scenarios 1 and 2 for the sensitivity analysis [47].

### 2.2.2. Value Functions

The values of the indicators are expressed in different units of measurement; to carry out the evaluation, it is necessary to homogenize all the values obtained in normalized values, this is done by means of value functions, which convert the physical units of measurement of each indicator into values from 0 to 1. From the different indicators, sustainability indexes are obtained, there being a value function for each indicator [45]. For each function, it will be necessary to establish degrees of preference, either maximum or minimum satisfaction, depending on the case for each indicator. In the definition related to the satisfaction of a given indicator, the MIVES method establishes a trend system, so that decreasing or increasing value functions can be considered, depending on what is considered as maximum or minimum satisfaction and allowing linear, concave, S-shaped, or convex value functions. In addition, it is necessary to add the mathematical expression that defines the value function, as expressed in Equation (1) [52].

$$V_{ind} = B \times \left[ 1 - e^{-\mathrm{k} \times \left( \frac{|X - S_{min}|}{C} \right)^{p}} \right] \tag{1}$$

The equation is interpreted as follows:

$V_{ind}$: corresponds to the value of the indicator being evaluated.

$B$: reference factor for the value function to be maintained in the ratio of 0 to 1. Therefore, it is understood that the maximum satisfaction ($X_{max}$) will have a value of 1, and the minimum satisfaction ($X_{min}$) will be 0. This factor is obtained by Equation (2).

$X$: is the abscissa that gives a value equal to $V_{ind}$.

*P*: defines the shape of the value function, concave, convex, linear, or S-shaped, for which the following values must be taken as reference: If *P* is less than 1, it corresponds to a concave curve; on the contrary, if *P* is greater than 1, the curve will be convex or S-shaped; finally, if *P* is equal to 1, the shape of the curve will be linear.

*C*: defines the *x*-value of the inflection point for curves with *P* greater than 1.

*K* defines the value of *y* at the point C.

$$B = \frac{1}{\left[1 - e^{-k \times \left(\frac{|S_{max} - S_{min}|}{C}\right)^P}\right]} \tag{2}$$

The parameters *P*, *C*, and *K* determine the shape of the value function. Figure 3 provides the shapes of the value function.

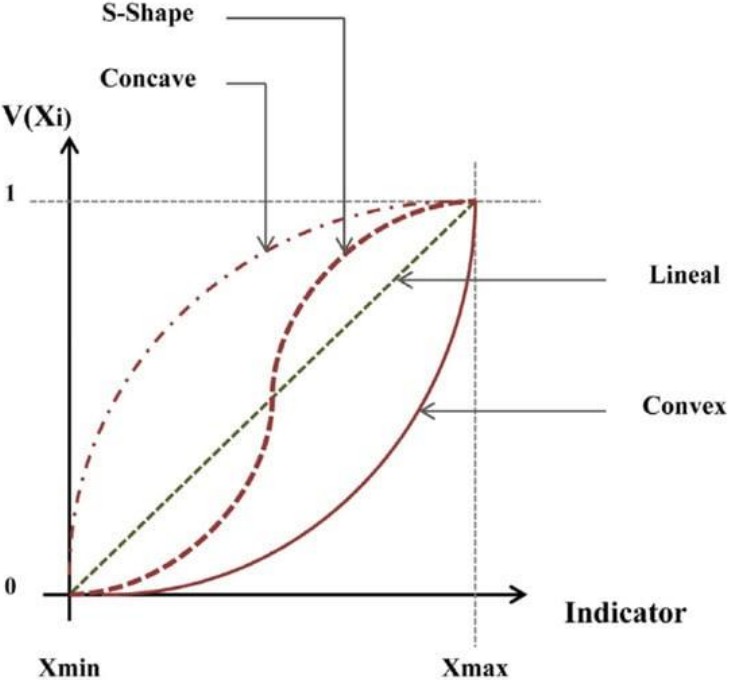

**Figure 3.** Forms of the value function [45].

According to the indicators presented in Table 1, the following value functions were adopted:

Indicators $I_1$, $I_2$, $I_4$, $I_6$, $I_7$, $I_8$, $I_9$, $I_{11}$, $I_{12}$ were modeled using increasing S-shaped value functions (SD). Satisfaction increases as favorable values are obtained, which are considered to be within a range of maximum satisfaction. In the case of indicator $I_3$, an S-shaped function was considered but with a decreasing trend since the satisfaction range for this indicator will be a decrease in the values obtained. The lower the value obtained, the higher the satisfaction, and therefore its trend will be decreasing. The indicators $I_5$, $I_{10}$, $I_{13}$ were modeled using linear increasing value functions (LC); a progressive increase in indicator values will mean higher satisfaction.

*2.3. Multicriteria Analysis*

2.3.1. Data Sources for the Evaluation

The indicators used in the evaluation were selected and quantified using databases and the participation of a panel of experts (Table 2), which provided the input data for the evaluation.

**Table 2.** Data from experts consulted.

| Area of Formation | Professional Field | Academic Degree |
|---|---|---|
| Business administration with emphasis on sustainable tourism | University academic | PhD |
| Ecotourism and ecotourism management | Independent workers, sustainable tourism consultants | Master's Degree |
| Sustainable territorial development | Formulation of social action projects | Master's degree |
| Sustainable tourism management | Executive director, Tourism Association | Master's degree |
| Biology and conservation | Formulation of conservation and environmental projects in NGOs | Master's degree |
| Business administration | Executive director of development association | Master's degree |
| Empirical formation | Tourism entrepreneur, hotel owner | Not applicable |
| Geographer and conservationist | University researcher | Master's degree |

Through the expert seminars, the different weightings for the decision tree were obtained, as well as the values used to make the base model of the evaluation and two scenarios of sensitivity analysis so that the validity of the method could be tested (Table 3). The sensitivity analysis allows us to know the influence of the different parameters on the value index obtained for each alternative [50]. This is achieved by varying the weights assigned to the requirements; variations in the weights at the level of criteria and indicators are not normally considered since the influence on the alternatives is often not significant [52].

**Table 3.** Assigned weights for the base model and sensitivity analysis.

| Requirement | Base Model | SA-Scenario 1 | SA-Scenario 2 |
|---|---|---|---|
| Environmental | 38% | 42% | 33% |
| Economic | 31% | 30% | 35% |
| Social | 31% | 28% | 32% |

2.3.2. Data Sources by Indicator

The study considered more than 90% of establishments that carry out activities related to tourism and, by default, the use of ES. In the Jimenez site, 80 establishments participated, representing 89%, while in Golfito, 76 establishments participated, representing 84% of the total. The indicators ($I_1$, $I_2$, $I_4$, $I_5$, $I_7$, $I_8$, $I_9$, $I_{10}$) were applied directly to each establishment. Indicator $I_3$, which refers to $CO_2$ emissions, was carried out as follows:

Inventory of vehicles used by hotels, tour operators, and restaurants for their tourism activities.

Questionnaire for people who offer cab services.

Questionnaire for boat owners.

In the case of air travel, the estimate was based on a reference study [60] and the frequency of monthly airline trips to Jiménez and Golfito.

Having referenced the stakeholders, an inventory was made of the different means of transportation, characterization, quantity, type of fuel, number of liters consumed monthly, and frequency of use; for the purposes of the analysis, it was estimated during the high season of visitation to the study site. The indicator $I_6$ complemented the application of questionnaires to the establishment and the consultation of official databases, in this case, those of the Costa Rican Tourism Institute.

The indicators $I_{11}$, $I_{12}$, and $I_{13}$ were applied in direct interviews with people from each site analyzed (150 in Golfito and 130 in Jiménez). For such purposes, a probabilistic sample

was considered, above, because all individuals have the same possibility of being chosen for the sample. For this, they are obtained according to the characteristics of the population and the sample size through a random or mechanical selection of the sampling units; in this case, the STATS program was implemented [61].

The questionnaires applied considered the following key elements:

Questionnaires to tourist establishments: These are divided into three parts, the first focuses on the environmental area, inquiring about aspects related to environmental management plans, environmental certifications, number of vehicles that the establishment has for tourism use, frequency of use, monthly fuel consumption, and most frequent destinations. The purpose of these questions was to obtain information on the environmental practices implemented by the establishment, as well as the data necessary to calculate the $CO_2$ emissions emitted as a result of transporting tourists to the establishment and places of interest within the tourist destination.

The second part of the questionnaire is made up of the economic component, which explores aspects related to whether the establishment has a tourism declaration, contracts with local suppliers as well as the percentage of products and services purchased locally, and how much is purchased monthly so that they can provide the necessary data to quantify elements that contribute to local economic development. Finally, it considered the social dimension, collecting data on the number of employees and how many of the total number of employees were local or local employees of the tourist destination as well as the nationality and owner of the establishment in order to identify aspects such as local entrepreneurship.

Social perception questionnaire: This questionnaire was applied randomly so that all individuals had the same probability of being chosen from the sample. The questionnaire asked about three key elements that responded to the perception indicators: level of belonging by the local community with respect to the ES, level of satisfaction with how the ES are used in tourism activities, and how the local population perceives the state and management of the conservation of the ES.

Questionnaire for persons offering land cab and maritime services: For both services, the main purpose of the questions was to quantify the amount of fuel used by their vehicles/boats monthly. For this purpose, it considered specifications of the type of fuel, type of vehicle/boat, most frequent destinations, and kilometers/nautical miles traveled per month. In this way, it was possible to obtain the data to calculate the $CO_2$ emissions emitted by land and marine transportation dedicated to tourism.

### 2.3.3. Calculation of Indicators

For indicators $I_1$, $I_2$, $I_4$, $I_5$, $I_9$, $I_{11}$, $I_{12}$, and $I_{13}$ a scoring scale of 1 to 5 or 1 to 3 was established, where 1 is the minimum satisfaction and 5 or 3 is the maximum possible satisfaction. For indicator $I_3$, once the information for the calculation was collected, it was performed based on the emission factors established by [62]. In the case of indicators $I_6$, $I_7$, $I_8$, and $I_{10}$ for the calculation, a weighting scale was established, where 0% is the minimum satisfaction and 100% is the maximum satisfaction; all these assignments are shown in Table 4.

For the application of the MIVES method, the maximum and minimum satisfaction ranges must be defined so that when integrating the data obtained from an indicator, there are limits as to which is the best or worst scenario. These values can be established according to the evaluator's own criteria, through consultation with experts, or by using databases of reference studies that indicate maximum or minimum satisfaction values. For the purposes of this study, the indicators were evaluated using a point scale, and percentage values were proposed by the author and presented to a panel of experts. The panel of experts also validated the characterization of the values assigned in the range of 1 to 5 or 1 to 3.

**Table 4.** Evaluation parameters and value functions assigned for the evaluation model. To interpret the table, C: approximates the abscissa of the inflection point, K: approximates the ordinate of the inflection point, P: is a shape factor that defines whether the curve is concave, convex, linear, or S-shaped, CS: S-shaped increasing function, DS: S-shaped decreasing function, CL: Linear increasing function, $X_{min}$ minimum satisfaction, $X_{max}$ maximum satisfaction.

| | Indicators | Units | Function | $X_{min}$ | $X_{max}$ | C | K | P | Jiménez | Golfito |
|---|---|---|---|---|---|---|---|---|---|---|
| $I_1$ | Environmental management and certifications | Points | CS | 1 | 5 | 3 | 0.5 | 3 | 3 | 2 |
| $I_2$ | Wastewater management | Points | CS | 1 | 3 | 2 | 0.5 | 3 | 2 | 2 |
| $I_3$ | $CO_2$ emissions | $TnCO_2$ | DS | 5057.03 | 0 | 2528.51 | 0.5 | 3 | 5057.03 | 3444.5 |
| $I_4$ | SE Management in Tourism | Points | CS | 1 | 3 | 2 | 0.5 | 3 | 2 | 1 |
| $I_5$ | Benefits of tourism for ecosystem services | Points | CL | 1 | 5 | 1.4 | 0.01 | 1 | 2 | 1 |
| $I_6$ | Tourist declaration | % | CS | 0 | 100 | 45 | 0.5 | 3 | 12.5 | 13.16 |
| $I_7$ | Procurement of local products and services | % | CS | 0 | 100 | 50 | 0.5 | 3 | 86.25 | 68.42 |
| $I_8$ | Benefits in the value chain | % | CS | 0 | 100 | 45 | 0.5 | 3 | 74 | 39 |
| $I_9$ | Jobs generated | Points | CS | 1 | 3 | 2 | 0.5 | 3 | 3 | 2 |
| $I_{10}$ | Local tourism businesses | % | CL | 0 | 100 | 10 | 0.01 | 1 | 53.75 | 68.42 |
| $I_{11}$ | Belonging to ecosystem services | Points | CS | 1 | 5 | 3 | 0.2 | 3 | 4 | 4 |
| $I_{12}$ | Perception of ecosystem services conservation | Points | CS | 1 | 5 | 3 | 0.5 | 3 | 1 | 1 |
| $I_{13}$ | Resident satisfaction with tourism | Points | CL | 1 | 5 | 1.4 | 0.01 | 1 | 3 | 3 |

## 3. Results

### 3.1. Calculation of Indicators and Value Functions

The calculation method used was by scoring from 1 to 5 or from 1 to 3, where 1 represents for all values the minimum satisfaction ($X_{min}$); on the other hand, 5 or 3 represents the ranges of maximum satisfaction ($X_{max}$). Weighting scales were also used, where 0 represents the minimum satisfaction and 100% the maximum satisfaction (except for indicator $I_3$), where the maximum satisfaction will be 0%. See Appendix A for the indicator-based estimation criteria.

In Table 4, the parameters used for the implementation of the model are shown, as well as the value function assigned for each indicator. The values obtained per indicator are shown according to the requirement in physical units.

The implementation of the model estimates the degree of satisfaction of each of the indicators and its value function through which the final values of each indicator are obtained in normalized units in a relation of 0 to 1 [47] which means that the closer it is to 1, the more sustainable it will be; on the contrary, the farther away it is, the less sustainable it will be. In Table 5, the normalized values obtained after applying the value function for each indicator are shown.

The results show low levels of sustainability, mainly in the environmental requirement indicators. The table above shows the results and the overall sustainability index of the two alternatives studied, showing that the Jiménez site presents better values than Golfito; however, both scenarios obtain very low sustainability indexes with respect to the ideal scenario. In Figure 4, the values can be observed at the level of the evaluated requirements (environmental, economic, social). The Y-axis shows the values obtained at the sustainability index level, while the X-axis shows the alternatives studied.

In the requirements, the Jiménez site has a better sustainability value than the Golfito site; however, the difference between one and the other is not significant. In addition, for both alternatives, the environmental requirement has the lowest value with respect to the others, with values of 0.027 for Golfito and 0.031 for Jiménez. The economic and social

requirements are the best for the Jimenez site with values of 0.19 and 0.18, respectively. The Golfito site has values of 0.11 for both requirements.

**Table 5.** Normalized values resulting from the application of the model according to the indicator.

|  | Indicators | Jiménez | Golfito |
| --- | --- | --- | --- |
| $I_1$ | Environmental management and certifications | 0.03 | 0 |
| $I_2$ | Wastewater management | 0.15 | 0.15 |
| $I_3$ | $CO_2$ emissions | 0 | 0.15 |
| $I_4$ | SE Management in tourism | 0.15 | 0 |
| $I_5$ | Benefits of tourism for ecosystem services | 0.5 | 0.25 |
| $I_6$ | Tourist declaration | 0.01 | 0.01 |
| $I_7$ | Procurement of local products and services | 0.94 | 0.74 |
| $I_8$ | Benefits in the value chain | 0.9 | 0.28 |
| $I_9$ | Jobs generated | 1 | 0.15 |
| $I_{10}$ | Local tourism businesses | 0.55 | 0.69 |
| $I_{11}$ | Belonging to ecosystem services | 0.48 | 0.48 |
| $I_{12}$ | Perception of ecosystem services conservation | 0.5 | 0.5 |
| $I_{13}$ | Resident satisfaction with tourism | 0.03 | 0.03 |
| $I_G$ | Global sustainability index | 0.40 | 0.25 |
| $SA_1$ | Sensitivity analysis-Scenario 1 | 0.38 | 0.24 |
| $SA_2$ | Sensitivity analysis-Scenario 2 | 0.43 | 0.27 |

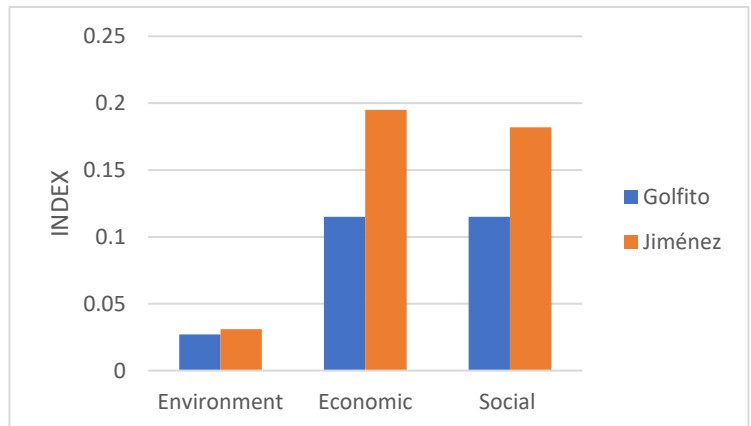

**Figure 4.** Sustainability values for both alternatives at the requirements level.

In Figure 5, the values obtained for both alternatives according to the evaluated indicator are shown:

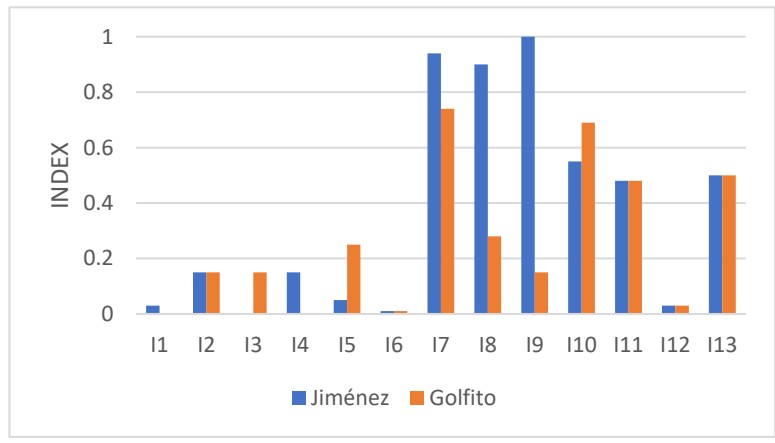

**Figure 5.** Values obtained by indicator for each alternative evaluated.

The results obtained at the indicator level reflect low levels of sustainability in the environmental indicators for both alternatives evaluated, with the environmental management plan and certifications ($I_1$), $CO_2$ emissions ($I_3$), and ES management ($I_4$) indicators the worst evaluated with values of 0.03 and 0.15. In general terms, the Jimenez site presents the best values; however, as a general reading, the difference in sustainability between both alternatives is not significant, presenting very similar values.

The economic and social requirements indicators are the ones that present the best sustainability values for both sites; with the acquisition of local products and services ($I_7$) and benefits in the value chain ($I_8$) indicators the best positioned in the evaluation, the Jiménez site is more favorable with respect to Golfito. The social requirements indicators are the most similar for both alternatives, with local tourism enterprises ($I_{10}$) and population satisfaction ($I_{13}$) being the best positioned.

### 3.2. Sensitivity Analysis

The sensitivity analysis is performed with the purpose of validating the results obtained in the initial model since there is no single alternative that is the best in each of the aspects evaluated; it allows us to know how much influence different parameters have on the value index obtained for each alternative. The sensitivity analysis is carried out by varying the weightings assigned to the requirements; variations in the weightings at the criteria and indicator level are not normally considered, since the influence on the alternatives is not usually significant [52].

The sensitivity analysis was carried out by assigning weights to the panel of experts, who were asked to consider two scenarios (Scenario 1 and Scenario 2), these values are shown in Table 3. For scenario 1, 42% was assigned to the environmental requirement, 30% to the economic requirement and 28% to the social requirement. For Scenario 2, 33% was assigned to the environmental requirement, 35% to the economic and 32% to the social.

The trend between both alternatives remains very similar, and no significant changes are observed between the sustainability values of the initial model with respect to the new scenarios, (see Table 5 maintaining the site of Jiménez as the most sustainable with respect to Golfito.

Figure 6 shows the variability of the global sustainability indexes for the three scenarios (initial model with respect to scenarios 1 and 2).

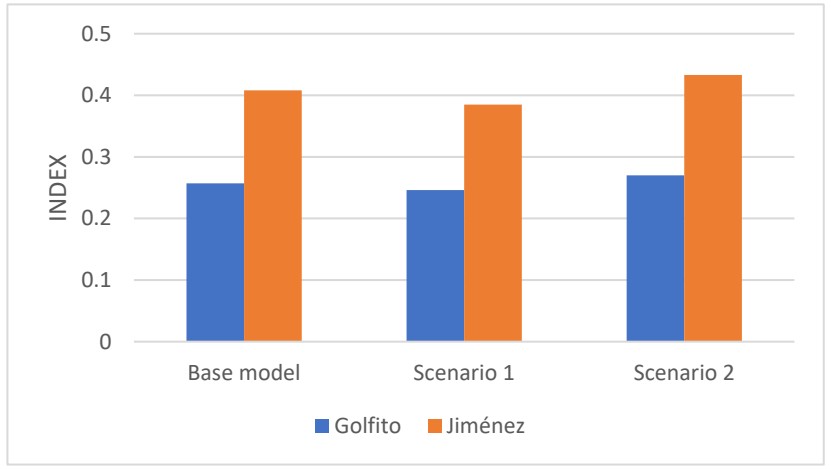

**Figure 6.** Global sustainability index for each scenario.

The variability of the sustainability indexes obtained for all scenarios is not significant, not exceeding 5% between the original model and scenarios 1 and 2 for both alternatives. It also allows for confirming the results obtained for the original scenario (evaluated), thus confirming the initial trend in which the Jimenez site remains the most sustainable alternative.

## 4. Discussion

Of the two alternatives studied with respect to tourism activities around the ES, the most sustainable is Jiménez over Golfito; however, the differences between one and the other are not significant, and both alternatives studied present low levels of sustainability. It can be observed that the global index does not even reach a score of 50%; Jiménez obtains 42% and Golfito around 28%.

To see the results obtained in more detail, through the requirements it can be seen how the lowest sustainability values correspond to environmental sustainability, with indicators $I_1$ (environmental management plan and certifications), wastewater management ($I_2$), $CO_2$ emissions ($I_3$), and SE management in tourism ($I_4$) being the worst evaluated. Under this scenario, it is important to point out that the majority of tourism establishments that maintain good environmental practices and therefore have some type of environmental certification; obtaining such recognition has a strategic purpose in order to take advantage of the country's promotional positioning as a "sustainable" country to attract environmentally responsible clients. On the other hand, smaller tourism establishments have limited access to this type of certification due to budgetary issues [63], and this behavior may affect the results obtained in terms of environmental requirement indicators since more small businesses benefit from tourism but do not have an environmental management program or maintain good environmental practices at a reduced level.

In the case of the $CO_2$ emissions indicator ($I_3$), the marked difference between the two alternatives is due to the distances traveled to reach the tourist attractions. For example, in Golfito, all the tourist attractions and activities are close to the distribution center, so fuel consumption is much lower because the distances are relatively short. In contrast, in Jiménez, where the main attraction is Corcovado National Park, the distance between the distribution center and the protected area is about a 50 km (100 km) round trip; therefore, fuel consumption will be much higher than in Golfito, directly implying a greater number of emissions from the transportation sector. The transportation sector is the main source of greenhouse gas emissions in Costa Rica; tourism is an activity that generates a high amount of travel and in the case of the site under study is no exception; it should also be noted that in the emission sources in the case of Jiménez and Golfito, being coastal sites, maritime activities contribute to increasing emission levels [60].

Regarding the ES management indicator in Tourism ($I_4$), its rating is low; however, values are only represented for the Jiménez site, which obtained a more favorable rating because about 46% of the total number of tourism establishments consulted, according to the rating scale, have an intensity of use of the ES in tourism activities with the implementation of conservation actions and therefore measure the impacts of their activity. The opposite is the case in Golfito, where 80% have an intensity of use without policies and actions for the protection and conservation of ES.

The wastewater management indicator ($I_2$) for both alternatives shows acceptable values in general terms. According to the proposed evaluation scale (1 to 3), both alternatives were placed on the scale at 2. This means that, although it is true that this is not the ideal scenario for wastewater treatment, at least there is evidence of treatment with septic tanks, which has fewer polluting implications than if the wastewater were discharged into the open, as is often the case.

The economic and social requirements presented better values at the Jiménez site compared to Golfito, highlighting indicators such as the purchase of local products and services ($I_7$) and benefits in the value chain ($I_5$). In the case of the first indicator, what is indicated is that of the total number of tourism establishments operating in the study area, 86.25% in Jiménez purchase local products and services, while only 68.42% in Golfito do so. For both alternatives, the values obtained are satisfactory, although there is a clear difference between Jiménez and Golfito. On the other hand, for the second indicator, ($I_5$), the Jimenez site presents better sustainability values (0.9) compared to Golfito (0.28). This trend is due to the fact that of the total number of tourist establishments, the Jiménez site

acquires a higher percentage of local products and services from the local community at 74%, with Golfito at 39%.

The tourism ES benefits indicator ($I_5$), the evaluation is related to the economic benefits generated by the activity for the population of the communities of the tourist destination. Of the two sites evaluated, Jiménez obtained better values than Golfito. According to the evaluation scale applied for this indicator, (See Table A1) the Jimenez site presents a better level of income than Golfito; obtaining 53% which qualifies it as "Good". The income scale for Jimenez is between US$750 and US$1100 on average. These values show that the income received is above the minimum wage. [64]. However, there are important challenges to improve the distribution of the benefits generated by tourism, given that the activity is often concentrated in a small number of tourism establishments, leaving many communities on the margins, which become transit sites to the main attractions, in this case, Corcovado National Park, causing the perceived benefits to be insufficiently representative for most of the communities of the destination [65]. In the case of Golfito, it had a rating of 2, with 65% of the people surveyed indicating tourism income between USD 450 and USD 750; on the rating scale, this is rated as "bad" given that in some cases it may be below the minimum wage.

The tourism declaration indicator ($I_6$) for both alternatives corresponded to 12.5% for Jiménez and 13.16% for Golfito; obtaining very low sustainability values for both sites. According to the rating scale (Table A1) for both alternatives, their rating is considered very poor. This may be because although there is a high-quality offer with the standards required by the tourism declaration, there are a greater number of low-category establishments, which are generally small establishments that are not required to obtain a declaration of tourist interest due to their operation, in addition to the fact that it would not be profitable given their characteristics.

Finally, the social requirement maintains the same trend with respect to the values obtained in the previous scenarios, with the Jiménez site having better values (0.18) compared to Golfito (0.11). The indicators with the greatest weight in the evaluation correspond to local tourism businesses ($I_{10}$) and community ownership of the SE ($I_{11}$). In the case of the jobs generated indicator ($I_9$), as shown in Figure 5, the best evaluation is for Jiménez, and according to the weighting scale, Jiménez obtained the highest satisfaction, that is, the jobs generated in tourism and through the use of the ES are a high percentage of permanent jobs for local people or people from the community, and they are also full-time jobs regardless of the tourist season. In the 0 to 1 estimate for this indicator, Jiménez obtained the ideal sustainability scenario, while Golfito scored only 0.15. This trend can be explained by the fact that Golfito functions more as a distribution center for tourism activity, in addition to the fact that the type of tourist visiting Golfito may be different from that of Jiménez, generating a major variability in employment [66].

The local tourism business indicator ($I_{10}$) in Golfito shows better values than Jiménez; this is due to the fact that in Golfito, of the total number of tourist establishments analyzed, 68% belong to local businesses, while for Jiménez, 53% belong to local businesses. The two indicators that are most similar in terms of results are those of belonging to the ES ($I_{11}$) and satisfaction of the local population ($I_{13}$); for both alternatives, the values obtained in a 0 to 1 ratio are 0.48 and 0.50, respectively, reaching almost 50% of the unit value for each one. The opposite is true for the indicator perception of the conservation of ES ($I_{12}$); the results show that the local population is dissatisfied with the state and conservation of ecosystem services, which confirms the serious existing problems regarding the conservation of biodiversity in the natural areas where most tourism activities are carried out in both sites.

## 5. Conclusions

The study in question has estimated the degree of sustainability of tourism activities in two sites (Jiménez and Golfito). It estimates, on the one hand, how sustainable the activities carried out around the ES are, and on the other hand, the benefits perceived as a result of

the practice of tourism in these sites; the interrelationship between human beings and the different services offered by the environment is evaluated.

In general, of the two alternatives studied, the Jimenez site is more sustainable than its Golfito counterpart; however, for both sites, the sustainability values obtained are significantly low, not even reaching 50% of unity for both destination sites, with Jimenez being the closest. Of the requirements studied, the economic requirement is the best positioned in the evaluation, with the Jiménez site showing the best values compared to Golfito. With respect to representativeness, each alternative in the economic requirement represents 11.5% for Golfito and 19.5% for Jiménez, followed by the social requirement, which obtained 11.5% for Golfito and 18.2% for Jiménez. At the environmental level, it presents the lowest values of sustainability, with representativeness with respect to the unit of 2.7% for Golfito and 3.1% for Jiménez.

At the level of indicators, Jimenez proved to be the most sustainable, except for the $CO_2$ emissions indicator ($I_3$), where Golfito comes out with better values (0.15). The indicator purchase of local products and services ($I_7$) throughout the analysis was the one that presented the best sustainability values for both alternatives, being the closest to an ideal sustainability scenario. Through sensitivity analysis, the weights of the requirements were varied, but in both scenarios the Jimenez site remained more sustainable than the similar Golfito site, with a variation that did not exceed 5% of the values obtained in the original case. Therefore, it is possible to conclude that the method is functional and that the results remain consistent in the new scenarios compared to the initial model.

The results of the study present a general framework for the examination of the sustainability of tourism activities related to the ES of the tourist destination Golfito/Jiménez, which may vary depending on the approach, data sources, method, and type of indicators to be used for the analysis. However, the results of the study indicate a trend in tourism activity in the destination and its relationship with the ES; it can serve as a tool for decision-making by the competent authorities.

The study and methodology applied are considered novel for evaluating ES; generally, evaluations are carried out from a more economic approach; it is unusual to find the integration of the three areas of sustainability in similar studies. Finally, the MIVES method has been applied even though its implementation is mainly in the field of engineering and construction, which makes it interesting since the exercise carried out confirms that the method can be very versatile and its field of action should not be restricted to only one; on the contrary, its application can occur in multiple scenarios and in the end it will provide data of great importance for decision-making in sustainability evaluations.

### 5.1. Limitations of the Research

Extrapolating the methodology to other areas outside Costa Rica may entail difficulties in terms of data availability for certain indicators; although in general terms there is a lot of information available, care should be taken in the selection of the information so that the judgments that may be made after the evaluation are well founded and transmit reliability.

Special care should be taken in the selection of the experts to be consulted, given that all areas of economic, social, and environmental knowledge should be represented, avoiding as much as possible that weightings of greater or lesser weight are suggested depending on the affinity for a specific area; for example, if only environmental experts are consulted, it is possible that they will give greater weight to the environmental requirement and so on with the other areas.

### 5.2. Future Research Directions

It has been proven that the method is flexible and can accept a diversity of topics to be evaluated in the field of sustainability, so it could be considered to apply the method for the evaluation and management of projects such as "poles" of tourism development. In addition, combining the method with life cycle analysis can be applied to evaluate how sustainable a certain material or product can be applied in certain constructions in

vulnerable sites, for example, in protected natural areas. Another scenario is to consider the application of the methodology in beach quality assessments so that decisions can be made regarding the ability of a beach to maintain its activity over time.

**Author Contributions:** Conceptualization, J.D.A.; methodology, J.D.A.; software, J.D.A.; validation, A.H., R.T. and J.V.; formal analysis, J.D.A.; investigation, J.D.A.; resources, J.D.A.; data curation, J.D.A.; writing—original draft preparation, J.D.A.; writing—review and editing, A.H.; visualization, A.H.; R.T. and J.V. supervision, A.H.; project administration, J.D.A.; funding acquisition, J.D.A. All authors have read and agreed to the published version of the manuscript.

**Funding:** This research has been financed by the teacher formation program of the University of Costa Rica through the granting of scholarships for study abroad. Scholarship Contract No. 24-2022.

**Data Availability Statement:** Not applicable.

**Acknowledgments:** As part of my doctoral studies, I have written this paper in response to the objectives of the research plan, for which I have had direction and tutoring by Ana Hernando and Rosario Tejera, as well as the collaboration and joint work with Javier Velasquez. This article contains part of the methodology of the final master's thesis project "Model for decision making in the evaluation of tourist destinations based on sustainability criteria: A case study located in Costa Rica". The work carried out is part of the scholarship program of the University of Costa Rica for the formation of professors abroad, through which the current doctoral studies in Engineering and Management of the Natural Environment at the Polytechnic University of Madrid are financed. We would also like to thank the panel of experts who participated in the selection and weighting of indicators for the final evaluation as well as the chambers of tourism of Jiménez and Golfito, tourism entrepreneurs, and institutions linked to tourism and the environment of Costa Rica.

**Conflicts of Interest:** The authors declare no conflict of interest.

## Appendix A

**Table A1.** Criteria and scales to estimate for each indicator.

| Indicators | Evaluation | Description |
|---|---|---|
| $I_1$ | 1 = Deficient<br>2 = Regular<br>3 = Good<br>4 = Very Good<br>5 = Excellent | If you do not have an environmental management plan (EMP) or certifications<br>If you only have an EMP<br>It does have an EMP and other types of environmental certifications<br>If you have both an EMP and a basic level of tourism sustainability certification<br>EMP and at the same time it has an elite level tourism sustainability certification |
| $I_2$ | 1 = Deficient<br>2 = Satisfactory<br>3 = Very satisfactory | Free discharge<br>Septic tank<br>Treatment plant |
| $I_3$ | $TnCO_2$ | Maximum satisfaction will be zero $CO_2$ emissions<br>Minimum satisfaction will be the maximum value obtained in the calculation of $CO_2$ emissions of one of the alternatives studied. |
| $I_4$ | 1 = Unsatisfactory<br>2 = Regular<br>3 = Very satisfactory | Intensity of use without protection and conservation policies<br>They carry out a regular intensity of use and implement partial conservation policies without measuring the possible impacts.<br>Implementation of protective policies and conservation |
| $I_5$ | 1 = Very bad<br>2 = Bad<br>3 = Good<br>4 = Very Good<br>5 = Excellent | Income less than USD 450<br>Between USD 450 and USD 750<br>Between USD 750 and USD 1100<br>Between USD 1100 and USD 1500<br>More than USD 1500 |

**Table A1.** *Cont.*

| Indicators | Evaluation | Description |
|---|---|---|
| $I_6$ | 1 = Very deficient<br>2 = Deficient<br>3 = Good<br>4 = Very good<br>5 = Excellent | Less than 30% of the establishments have been declared as tourist destinations.<br>Greater than or equal to 30% of the establishments with a tourism declaration<br>More than 50% of the establishments with tourist declaration<br>More than 70% of the establishments with tourist declaration<br>More than 90% of the establishments have been declared as tourist destinations |
| $I_7$ - $I_8$ - $I_{10}$ | 1= Very deficient<br>2 = Deficient<br>3 = Regular<br>4 = Very Good<br>5 = Excellent | 0 to 20% Unsatisfactory<br>21 to 40% not satisfactory<br>41 to 60%<br>61 to 80% Satisfactory<br>81 to 100% Very satisfactory |
| $I_9$ | 1 = Deficient<br>2 = Regular<br>3= Very satisfactory | Full-time permanent jobs, but not belonging to locals<br>Local permanent jobs, but not full time<br>Jobs that are permanent, local and full-time |
| $I_{11}$ - $I_{12}$ - $I_{13}$ | 1<br>2<br>3<br>4<br>5 | Very poor and unsatisfactory<br>Deficient and unsatisfactory<br>Regular<br>Very good and satisfactory<br>Excellent and very satisfactory |

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
