# Peer review of "Sustainable Tourism around Ecosystem Services: Application to a Case in Costa Rica Using Multi-Criteria Methods"

_land, doi:10.3390/land12030628_

Round 1

Reviewer 1 Report

Your work is a novel approach to a subject that has been treated from many perspectives. It is therefore a courageous approach, but one that entails major difficulties. I would like to make the following clarifications:

- In my opinion, you should study the concept of sustainability in greater depth. It is not uncommon to find that sustainability is treated in a similar way to competitiveness. And these two concepts are not synonymous. This boils down to talking about "more sustainability" and interpreting systems and indicators in this sense. It would be very interesting if you could address this issue.

- A more thorough explanation of the suitability of the method and its applicability is very timely. Why has a similar approach not been used before? Point out the key issues.

- In section 2.3.2, it would be useful to detail the questionnaires discussed in order to understand the construction of the indicators.

- You should explain in more detail the construction of the indicators (explanatory diagrams are always welcome).

- In section 2.3.3 it is also useful to detail the reasons for the choice of indicator scales.

- If necessary, include appropriate annexes.

- In my opinion, I would reinforce the conclusions with a more generalisable approach to your research. Look for a more conceptual level to take advantage of the novelty of your work.

I hope that the above will help you to strengthen your paper.

Reviewer 2 Report

Greetings,

The paper is well written. All selections are well written. It is necessary to make certain corrections. In the abstract, it is necessary to emphasize the most important results of the research. Hide numbers in keywords. In the introduction, it is necessary to emphasize the aim of the research and the contribution of the research. In selection 2. Materials and Methods, it is necessary to explain how the percentages for the criteria were obtained. It is still necessary to explain how the scenarios were formed. In the discussion, it is necessary to provide references and link them to the obtained results. In the Conclusion, it is necessary to state the limitations of the research and directions for future research.

All the best.
